# Consent and Complications in Health Care: The Italian Context

**DOI:** 10.3390/healthcare11030360

**Published:** 2023-01-27

**Authors:** Maricla Marrone, Enrica Macorano, Giuseppe Lippolis, Pierluigi Caricato, Gerardo Cazzato, Antonio Oliva, Benedetta Pia De Luca

**Affiliations:** 1Section of Legal Medicine, Interdisciplinary Department of Medicine, University of Bari “Aldo Moro”, 70124 Bari, Italy; 2Andrology and Kidney Transplantation Unit, Department of Emergency and Organ Transplantation-Urology, University of Bari, 70124 Bari, Italy; 3Section of Pathology, Department of Emergency and Organ Transplantation (DETO), University of Bari “Aldo Moro”, 70124 Bari, Italy; 4Department of Health Surveillance and Bioethics, Section of Legal Medicine, Catholic University, Fondazione Policlinico “A. Gemelli” IRCCS, 00100 Rome, Italy

**Keywords:** informed consent, urinary catheter, nursing procedure

## Abstract

Informed consent is the manifestation of the will that a patient freely expresses toward a medical treatment. The physician is responsible for acquiring informed consent for both medical and nursing procedures. Informed consent represents a juridical–deontological tool that allows therapeutic choices to be shared with the user after having exhaustively explained the risks and benefits of the procedure itself. In fact, the physician has an obligation to provide the patient with clear and comprehensible information about the type of service, the methods of delivery, the benefits, the risks, even unforeseeable ones, and the complications. According to Italian legal guidelines, in cases of presumed health responsibility, the health professional accused of negligence will have to demonstrate that any complication that has arisen, although foreseeable, was not preventable. Through the analysis of a clinical case relating to the procedure of insertion of a bladder catheter performed by a nurse and a review of the literature, the authors explain the importance of the information that must be provided to the patient before carrying out any invasive procedure, even if not performed by the doctor. The authors describe the problem in the Italian context and propose a possible solution.

## 1. Introduction

For the Italian legal orientation, the concept of “complication” is unnecessary: in cases of alleged malpractice liability, it is not enough to prove the occurrence of complications in the course of acts and procedures on the patient. The practitioner has to prove that his failure to perform was brought about by an additional cause not attributable to him. In other words, he must prove that he was unable to perform because of an event that, although foreseeable (complication), was not avoidable.

Italian physicians must therefore be able to demonstrate that if a complication occurs, it is the consequence of an unforeseeable event and not an error.

It is, therefore, not enough that it is a foreseeable complication, but proof is required from the doctor that everything that had to be done to prevent the complication from occurring was done.

The complication must be the consequence of “fate” and not the doctor’s actions.

The doctor, therefore, even if he follows the guidelines during his activity, but cannot prove it, could be found guilty of error by Italian law.

In addition, a further aspect to consider is that unforeseeable and unpreventable complications do not turn out to be grounds for censure for the health care providers only if the patient, properly informed by them, has agreed to undergo the health care procedure and its risks. If, on the other hand, there is no valid form of consent from the patient, such a complication becomes grounds for censure for the health care provider since there is no evidence that the health care provider acted according to the patient’s wishes.

Again, if the patient had not been informed of the complication in question, even if it occurs due to an unpreventable cause, the health care provider may have to “pay” the share of “damages” caused by the violation of the right of self-determination, a direct consequence of the failure to provide informed consent. 

The sterile closed-loop bladder catheter placement procedure involves inserting a latex or silicone drain into the bladder through the urethra. It is an external urinary drainage system in a closed bag. Such a drain is also equipped with a urine sampling device and a tap attached to the bag itself that allows it to be emptied periodically without having to interrupt the circuit. This has made it possible to adequately prevent the occurrence of urinary tract infections [1].

The bladder catheter can be left in place for less than 7 days (short-term bladder catheter), between 7 and 28 days (medium-term bladder catheter), for more than 28 days (long-term bladder catheter), and for long periods (permanent bladder catheter) [2].

The bladder catheter insertion procedure is generally a nursing procedure and involves several consequential steps. It is necessary to specify that patients may be subjected to undergoing:₋short-term bladder catheterization;₋long-term bladder catheterization;₋intermittent bladder catheterization.

In any case, it is essential that operators are trained in the insertion and management of the bladder catheter and that catheterization and all the maneuvers performed on the catheter are performed only by qualified personnel. All this is in order to prevent any complications that may be caused by the long stay of the implanted catheter.

In particular, the nurse must first inform the patient in order to obtain his cooperation and consent, perform antiseptic hand washing, prepare the necessary materials for intimate hygiene and insertion procedure, place the screen in order to ensure privacy for the patient, and perform personal hygiene for the patient if the patient is not self-sufficient. Then, opening the kit package while maintaining asepsis, setting the sterile field, and following the catheter insertion procedure [3,4].

This procedure, even if performed routinely, is not without complications. In fact, the possible events could be traumatic injuries, urinary tract infections, bacteriemia secondary to the departure from the bladder catheter, and urethral lacerations or the creation of false pathways. For this reason, bladder catheters should be inserted only when there is a clear clinical indication and should be removed as soon as possible as the indication of use cases, in particular in the case of [5,6]: ₋acute urinary tract obstruction and urinary retention;₋permanent neurological bladder dysfunction;₋diuresis monitoring in critically ill patients or candidates for major surgery;₋surgery requiring an empty bladder;₋treatment of bladder neoplasms with topical cytotoxic drugs and brachytherapy of prostate cancer;₋performance of bladder function tests;₋emptying of the bladder before delivery, where the patient may rarely urinate spontaneously; ₋urinary incontinence; ₋severe cases of macrohematuria and pyuria.

The case report presented below concerns a very rare complication that can occur during the catheter insertion procedure and, specifically, ureter injury. 

Injury to the ureter is usually iatrogenic or from external trauma. Iatrogenic injury is observed after urologic (42%), gynecologic (34%), and general surgical (24%) procedures, with the vast majority of injuries observed at the distal ureter (91%) [7,8].

The most frequent complications of Foley catheter placement are urethral trauma and retention of the Foley balloon in the urethra.

Inadvertent placement of a Foley balloon within the ureter is a rare complication of urethral catheterization, with only seven cases reported in the medical literature (three right, four left) with variable clinical presentation [9,10,11,12,13,14,15,16]. 

The main purpose of the study is to analyze possible issues related to consent regarding all medical and nursing procedures. 

In the Italian context of constitutional principles, the signing of a valid informed consent constitutes the expression of the principle of inviolability of human freedom (Art. 13 of the Italian Constitution), from which the right of self-determination with respect to one’s own body derives. A fundamental issue relates to the mode of relationship between practitioner and recipient of the service, particularly between physician and patient.

The common practice in Italy is the acquisition of written informed consent for invasive practices of “medical” relevance. Informed consent must be signed by the patient in the presence of the doctor, who must clearly, comprehensively, and effectively expose all possible risks related to the execution of the medical practice, such as surgery.

In contrast, bladder catheter placement is a routine invasive nursing practice.

The issue proposed in the case analyzed thus raises some questions. In particular, do nursing procedures, such as bladder catheter placement, require informed consent? Does a routine nursing procedure require informed consent to be signed each time the patient goes to the hospital facility to undergo that particular procedure? Or is an informed consent signed only at the beginning of said sessions sufficient?

### Case Report

A 63-year-old man with chronic renal failure stage V K-DOQI under periodic/weekly hemodialytic replacement treatment, with neurogenic bowel and bladder due to outcomes of polio contracted in childhood, chronic obstructive pulmonary disease (COPD), arterial hypertension, hypothyroidism, previous pulmonary embolism undergoing drug treatment with an oral anticoagulant, underwent periodic replacement of the permanent bladder catheter (day 1) with Foley 16 Ch. No informed consent was signed during this procedure; furthermore, no noteworthy issues and/or elements were found. Three days later, the patient began to experience respiratory dyspnea, so he accessed the emergency department of the nearest hospital (day 3). On radiological investigation, a bronchopneumonic focus was found, so the patient was admitted and started on drug therapy. 

On admission to the inpatient ward (day 4), mild hematuria was detected, as well as significant neutrophilic leukocytosis, which could well correlate with the bronchopneumonic outbreak diagnosed in the patient [(leukocytes 29.69 × 103 µL (v.n. 3.70–9.70), neutrophils 92.9% and of Protein-C-reactive Protein (CRP) 222 mg/L (v.n. < 2.9)]. An anemic state was also evidenced, traced to the condition of chronic renal failure, treated with an erythropoietin analog and certain hemotransfusions. On day 7, a major contraction of diuresis was appreciated, noting the presence of 25 mL/24 h of urine in the bladder catheter pouch.

During the course of the hospital stay, the patient underwent the planned hemodialysis sessions according to his triweekly therapy schedule. The patient had no algic symptoms except for transient abdominal pain (day 7) that regressed following the performance of an evacuative enema.

In addition to this, on the eighth day of hospitalization (day 8), the suspicion of an entero-cutaneous fistula was raised, for which instrumental investigations (abdomen-pelvis CT scan) were requested, which confirmed its presence in the gluteal region. The abdomen-pelvis CT scan performed (day 10) showed, however, an additional interesting finding: at the right iuxta-vesical ureter, the presence of the apex of the bladder catheter was evident. Therefore, we proceeded with performing pyelo-CT, which revealed a fissure in the tract upstream of the catheter apex, extending for an approximately 3 cm segment of the mid-distal ureter, and contrast medium spread in the right pararenal space. 

There was also evidence of concomitant structural subversion of the ipsilateral kidney and perirenal soft tissues due to the presence of multiple abscess collections with concomitant hydroaerial levels. Therefore, the patient underwent emergency right nephrectomy surgery (day 10). Histological investigations performed on the operative piece confirmed the diagnosis of chronic active erosive-ulcerative phlogosis in abscessal evolution, associated with granulation tissue of the excretory tract, extending to the renal parenchyma and perirenal tissues. The patient died 3 days after (day 13) surgery due to septic status.

Figure 1 shows the timeline of this case.

## 2. Materials and Methods

A systematic review was elaborated following the preferred reporting items for systematic reviews and meta-analyses (PRISMA) guidelines.

The three different databases (PubMed, Google Scholar, and Web of Science) were consulted, using the main keywords “informed consent” crossed with the terms “urinary catheter”, “nursing procedure”, and/or “complications”. The research aimed to understand the reflections already posed on the topic.

The selection of the articles was carried out through the evaluation of both the title and the abstract.

The inclusion criteria were:₋Articles published in English;₋Type of paper: Original article, research article, systematic review, review.

The selected documents that met the inclusion criteria were then reviewed, as well as their references. 

## 3. Results

Among all the articles analyzed, the following types were found:₋Article: 4;₋Review: 3;₋Critical trial/Experimental study: 1;₋Supplement: 1;₋Communication: 3;₋Research article: 2;₋Case report: 7;₋Guideline: 1.

Of the 22 isolated and analyzed articles, the majority (9 papers, 40.9%) came from the United States, while 4 articles (18.1%) came from the United Kingdom, 3 articles (13.6%) came from Europe, 2 articles (9%) from Japan, 2 articles (9%) from Korea, 1 article (4.5%) from Australia, and 1 article (4.5%) from Mexico (Figure 2).

## 4. Discussion

Informed consent is the manifestation of will that the patient freely expresses regarding a health treatment. In other words, it represents a legal–deontological instrument that allows the health care professional to share treatment choices with the user. 

An important point of reference is the “Convention on Human Rights and Biomedicine”, drafted on 4 April 1997 in Oviedo and ratified in Italy in 2001 by law number 145.

The concept of consent is set out in Chapter II, Article 5: “An intervention in the field of health cannot be carried out unless the person concerned has given free and informed consent. This person shall first receive adequate information about the purpose and nature of the intervention and its consequences and risks. The person concerned may, at any time, freely withdraw his or her consent”.

This concept was further reinforced by the “Universal Declaration of Bioethics and Human Rights,” enshrined by UNESCO on 19 October 2005, which states in Article 6:

“*Any preventive, diagnostic or therapeutic medical intervention must be carried out with the prior free and informed consent of the person concerned, based on adequate information. Consent, where appropriate, must be given and may be withdrawn by the person concerned at any time and for any reason, without consequent disadvantage or prejudice*.”

To be considered valid, informed consent must meet certain requirements, namely being:₋Personal: expressed directly by the subject for whom the assessment is intended, except in cases of incapacity, concerning minors and the mentally ill;₋Free: not conditioned by psychological pressure from other parties;₋Explicit: manifested clearly and unambiguously;₋Informed: formed only after the patient has received all the information necessary to make a decision;₋Specific: in the case of particularly complex treatment, the patient’s acceptance must be directed toward such procedures, whereas an entirely generic consent to treatment would have no legal value. In some special situations, such as those related to a surgical procedure in case there was uncertainty about the degree of expansion and invasion of a neoplasm, expanded consent is used;₋Current;₋Revocable at any time.

The nurse usually intervenes between the doctor and the patient, facilitating the acquisition of information and acting as a guarantor of effective communication.

They are responsible for supporting the person to be treated by assisting the doctor, ensuring that he or she is fully informed about his or her abilities, autonomy, and treatment plans [17]. 

As regards purely nursing procedures, such as the case report on the procedure for inserting the bladder catheter, the nurse has independent information competence regarding general nursing care related to the analysis of the health needs of the person, nursing diagnosis, nursing objectives, and evaluation of the results achieved [18].

However, the problem that sometimes arises in relation to routine, even invasive, nursing procedures is that implicit and unwritten consent is adopted [19,20]. 

The obligation for the health care provider to have a valid informed consent signed by the patient/assisted person is reflected in the Italian Constitution; in particular, Article 13 enshrines the inviolability of personal freedom, and Article 32 that states that “*no one can be obliged to a given health treatment except by provision of law*”.

However, with regard to the bladder catheter introduction procedure, there are no universal guidelines regarding the acquisition of consent and the healthcare figure who should be in charge of the procedure.

In 2009, the Healthcare Infection Control Practices Advisory Committee (HICPAC), an advisory committee of the Centers for Disease Control and Prevention (CDC) in Atlanta, published guidelines for the prevention of catheter-related infections in which guidance on appropriate and inappropriate bladder catheter placement was listed, but there was no mention of whether or not consent was acquired and how, if at all.

The list was based on a critical review of the scientific literature available at the time. Due to the lack of high-quality studies examining indications for urinary catheterization, the recommendations for catheter use primarily represented expert consensus opinion [21].

In May 2015, the recommendations were revised by other authors and published in a special supplement in the journal *Annals of Internal Medicine*. A panel of 15 experts updated the recommendations using the RAND/UCLA adequacy method, which combines scientific literature review and expert opinion on health topics to assess whether the expected benefits of a medical procedure outweigh the potential harms [22]. Again, the recommendations concerned technical procedures but never the acquisition of informed consent.

In the Italian context, there are important inter-regional dissimilarities regarding the procedure of bladder catheter insertion, as in some regions, there is a very precise regional regulation regarding the performance of this procedure, while in others, there is no regulation whatsoever. Moreover, since this is a routine procedure performed by nursing staff, there is often no acquisition of any kind of written informed consent.

For example, the Lazio Region in January 2022 drafted a guideline document on informed consent aimed at defining the essential elements that each healthcare facility must comply with following current regulations [23]. 

Taking into account what has just been stated, should the procedure of bladder catheter introduction, in the example of the case report we presented, be considered a routine procedure, and therefore, implied consent may possibly be accepted? Conversely, considering the risks and complications related to the said procedure (even the rarer ones), is it necessary to acquire “official” consent at the beginning of the treatment course and whenever the procedure is repeated?

Absurdly, until Law 217 of 2019 was enacted in Italy, which specifically enshrines how informed consent should be obtained in health care, effectively encouraging precise and targeted case-by-case disclosure, i.e., health act by health act, it was a well-established practice to have a generic and all-inclusive consent form signed upon admission. That meager form included all possible invasive and non-invasive procedures to be implemented on the patient.

While the enactment of the new legislation has favored more accurate patient information about specific treatments (surgical and anesthesiological), it has, in fact, left “uncovered” others perhaps considered of lesser importance.

In fact, according to Italian Supreme Court ruling III Civil Section, 18 December 2015–20 May 2016, in 10,414 cases, the patient’s failure to sign the Informed Consent allows the patient to claim compensation.

There are many sentences of the Court of Cassation that speak of “damage from omitted informed consent”; for example, Cass. civ. sez. III 12 May 2021, no. 12593.

In the aforementioned judgment, the Court of Cassation specifies that the breach by the doctor of his duty to inform the patient may cause two different types of damage: damage to health (if it can be proved that the patient would have avoided undergoing the intervention and suffering the disabling consequences thereof if correctly informed) and damage from impairment of self-determination (when, due to the lack of information, the patient has suffered an asset or non-patrimonial prejudice other than the impairment of the right to health) (cf. Cass. no. 11950/2013 and Cass. no. 28985/2019).

With specific reference to the hypothesis of a correctly performed medical practice, from which, however, harmful consequences for health have arisen, where such intervention has not been preceded by adequate information to the patient regarding the possible prejudicial effects, the doctor may be called upon to pay compensation for the damage to health only if the patient proves, also by means of presumptions, that, if fully informed, he would probably have refused the intervention (Court of Cassation no. 2847/2010 and Court of Cassation no. 2998/2016; more recently, Court of Cassation no. 7248/2018).

Failure to provide informed consent violates the patient’s right to self-determination; therefore, the patient should be compensated regardless of whether the surgery or health service to which he was subjected was successful or not. 

In fact, the health care provider has an obligation to provide the patient with clear and intelligible information about the type of service, mode of performance, benefits, and risks, even unforeseeable risks and complications; he or she must make sure that the patient has understood the content of the information and make himself or herself available for any doubts, then proceed to sign the informed consent form.

Whenever this does not happen, the healthcare provider and the healthcare establishment in which it operates may be ordered to pay damages regardless of the success of the procedure. Thus, even if the service provided did not cause any harm to the patient and was conducted very well, including the resulting benefits.

It follows that the damage for failure to provide informed consent is independent of the damage to health. According to the Supreme Court, if damage to health also arises, such compensation is in addition to that for failure to consent; they are, therefore, two different torts, giving rise to two different compensations.

Therefore, in the Italian legal system, it is a widely shared principle that informed consent constitutes a rule of legitimacy and foundation of medical treatment, so much so even before Law No. 219 of 2017 came into force in Italy [24,25]. 

In the European context, the principle of informed and free consent to medical treatment tends to be progressively consolidated in jurisprudence. In particular, the Strasbourg Court has held that failure to comply with the requirement of informed consent violates Article 8 of the European Convention on Human Rights on the fundamental right to respect for private life: 

“*1. Everyone has the right to respect for his private and family life, his home and his correspondence. 2. There shall be no interference by a public authority with the exercise of this right except such as is in accordance with the law and is necessary in a democratic society in the interests of national security, public safety or the economic well-being of the country, for the prevention of disorder or crime, for the protection of health or morals, or for the protection of the rights and freedoms of others*”.[26]

In addition, the principle of informed consent results in a judicial problem, which of course, also involves, in addition to civil liability, criminal liability. In Europe, most legal systems do not provide for an ad hoc case aimed at criminalizing an intervention that differs partially or totally from the content of the consent given by the patient. Portugal and Austria are exceptions.

In fact, Article 156 of the Portuguese Criminal Code of 1983 punishes, with imprisonment of up to three years, a physician who acts without acquiring the informed consent of the patient. Similarly, Art. 110 of the Austrian “Strafgesetzbuch” enshrines the punishability of arbitrary medical treatment by stating, “*anyone who treats another person without his consent even if he acts in accordance with the rules of medical science shall be punished by imprisonment of up to six months*”.

Even in light of all this, relative to the case report presented by the authors, a rare complication occurred following the catheter insertion procedure that resulted in the patient’s death. It is true that the patient was undergoing continuous bladder catheter insertion procedures and had never presented any complication, but it is also true that this does not justify the failure of the health care providers to obtain the informed consent. Although considered a routine procedure for the patient, it is still invasive, and the risk/benefit ratio may vary over time. Therefore, the authors reiterate the usefulness of acquiring informed consent whenever the patient is to undergo an invasive procedure, even in the case of a procedure routinely performed on the patient. Perhaps in this specific case, since it is the same procedure repeated several times to the same patient, he could be considered adequately informed already after the first acquisition of consent. Therefore, a single and initial signing of informed consent may be sufficient.

## 5. Conclusions

In conclusion, the authors would like to take up paragraphs 8 and 10 of Article 1 of Law 219 of 2017, according to which, “*The time of communication between doctor and patient constitutes time of care”* and *“The initial and continuing education of physicians and other health professions includes training in patient relations and communication, pain management and palliative care*”.

It is essential for any type of healthcare procedure to acquire the informed consent of the patient through an “effective” communication.

According to Italian law, the relationship of care and trust between the patient and the doctor, which is based on informed consent, must therefore be promoted and valued. The text governs how this informed consent can be expressed: “the informed consent, acquired in the ways and with the instruments most suited to the patient’s condition, is documented in written form or through video recordings or, for the person with disabilities, through devices that allow him to communicate. Informed consent, in any form expressed, shall be entered in the medical record and in the electronic health record”.

Still, at any time, the person can review his health decisions. Refusal (not initiation) or renunciation (interruption) concerning all diagnostic tests and health treatments, among which the law includes artificial hydration and nutrition.

Cold and aseptically truthful information may reinforce the perhaps more burdensome but also more effective therapeutic choice and, in any case, may induce a more “participating and fighting” attitude of the patient; but, conversely, in other cases, it may lead to attitudes of resignation, depression, despair, even distrust of the health care provider or feelings of anger and resentment. Thus, an “accompaniment” to information is needed as an aspect or corollary of the complex doctor–patient relationship so that the patient has all the tools to be able to make decisions about his or her health status.

Therefore, the Authors in this paper reiterate the usefulness of acquiring informed consent when the patient has to undergo an invasive procedure. In the case of a procedure performed routinely, the patient could be considered adequately informed already after the first acquisition of informed consent. This concept can be considered valid both for the execution of nursing procedures and for the procedures put into practice by medical staff. Therefore, a single and initial signature of informed consent could be sufficient, but considering all the risks related to the procedure, using the concept of “expanded consent” is already in use for surgical interventions.

According to the concept of “expanded consent”, if the diagnostic investigations carried out before an operation do not allow the surgeon to have a definitive and certain prediction of the operation itself, the surgeon will be able to operate on the operating table on the basis of the situation that will arise once the intervention, without having to acquire further informed consent. In fact, it is possible to talk about “expanded consent”, for example, in cases where the diagnostic investigations performed before the surgery did not allow the surgeon to have a definitive and certain prediction of the surgery.

Similarly, for routine invasive procedures, “expanded consent” could be adopted. In this case, the patient would sign the consent only the first time he undergoes the aforementioned invasive procedure, expressing a valid consent also for the subsequent execution of the same procedure. Indeed, in such cases, the invasive procedure is repeated several times on the patient, always exposing him to the same risks and complications.

Even in this case, however, the possibility of revoking the consent given would be valid. Although the patient has already given consent to the first treatment, he or she may, in the course of subsequent treatments, decide to revoke it and not to undergo the same treatment again. 

In this case, according to Italian law, if the patient expresses the renunciation or refusal of health treatments necessary for his survival, the doctor must propose to the patient and, if he consents, to his family members, the consequences of this decision and possible alternatives and promote any action to support the patient himself, including by making use of psychological assistance services. 

In any case, the doctor is obliged to respect the patient’s expressed desire to refuse treatment or to renounce it and, as a result, is exempt from civil or criminal liability. On the other hand, the patient cannot demand medical treatment contrary to the law, professional ethics, or good clinical-care practices; in the face of such requests, the doctor has no professional obligations.

As with surgical practices, the patient would thus be free to self-determine his or her own health, but this freedom would not slow down the treatment process: the health professional would be protected by law even if he or she did not have to have written proof of consent for each time he or she performed the same procedure.

## Figures and Tables

**Figure 1 healthcare-11-00360-f001:**
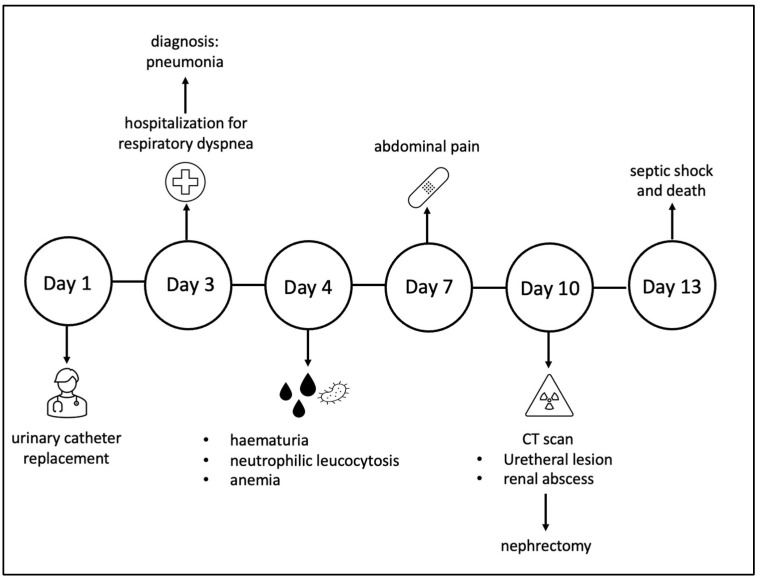
The timeline of this case.

**Figure 2 healthcare-11-00360-f002:**
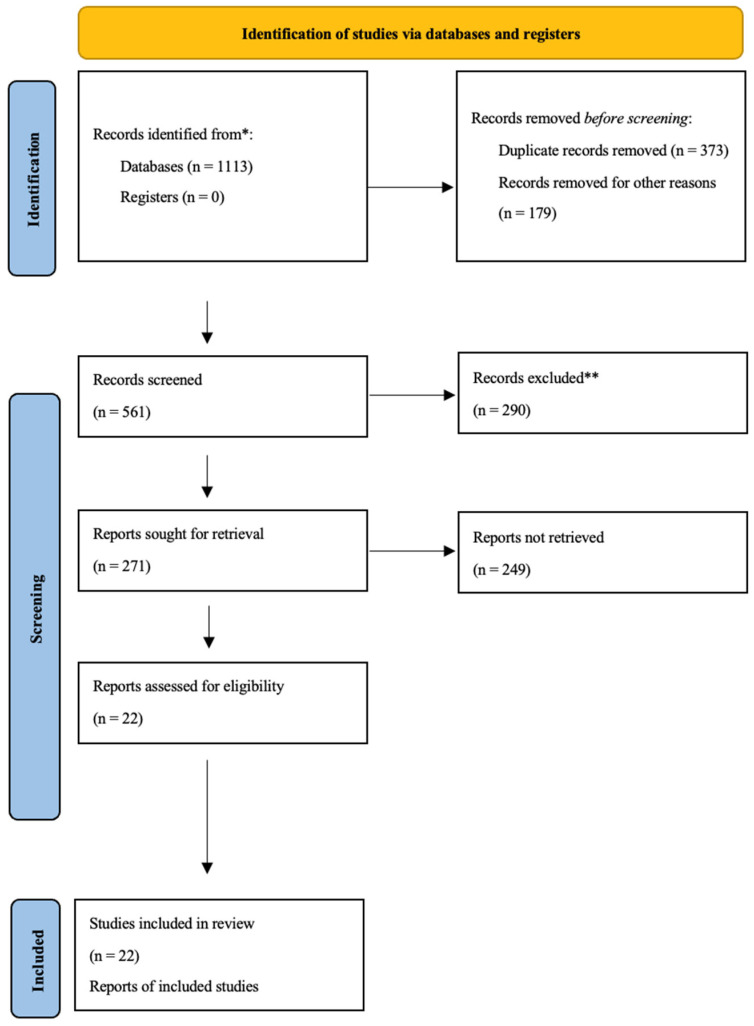
PRISMA guidelines used in a review of the literature. * The number of records identified from each database or registry searched is reported. ** No automation tools were used to obtain this figure; the records shown were excluded by a human operator.

## Data Availability

Not applicable.

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
