# Peer review of "Consent and Complications in Health Care: The Italian Context"

_healthcare, 2023, doi:10.3390/healthcare11030360_

Round 1

Reviewer 1 Report

The article has seven coauthors.

They write in the abstract that "Through the analysis of an interesting clinical case relating to the procedure of insertion of the bladder catheter performed by the nurse and a review of the literature, the authors explain the importance of the information that must be provided to the patient before carrying out any invasive procedure, even if not performed by the doctor". Given the unfortunate outcome that this case had (the only one analyzed, it is worth mentioning), namely with the death of the patient under analysis, 63 years old, I would suggest that the word "Interesting" be removed.

In the Introduction, the authors refer to Italian legal guidelines on this context, but only in the "Discussion" part do they allude to the regulations in question, as well as the manifest insufficiencies. It would be appropriate to anticipate this legal context already in the "Introduction", but based on the main references.

The authors explained in detail the clinical situation of the patient and even made an illustrative timeline for a faster understanding. They also performed a systematic review, prepared according to the reporting items for systematic reviews and meta-analyses (PRISMA) guidelines. A schematic outline of these procedures was also made. Three different databases (PubMed, Google Scholar, and Web of Science) were consulted.

The "Discussion" part is well written and provides a comprehensive explanation of the clinical, legal, and regulatory contexts surrounding these cases. It is the most successful part of the work, followed by the clinical description and the evolution of the patient analyzed.

The "Conclusions", however, seem to fall short of expectations, with only three paragraphs, the first of which is restricted to direct quotes. As there are seven co-authors, and after analyzing a clinical case that deserves care, especially because it urges that these conditions be the subject of increased vigilance, I would have expected a joint opinion from the authors on development strategies to be proposed and implemented for the benefit of patients and the clinical community. 

The article could be improved, as pointed out above: by a more robust "Introduction" based on essential references; by more extensive "Conclusions" linked to the authors' opinion that, as an epilogue, outline what can be done to improve this clinical context. 

Author Response

They write in the abstract that "Through the analysis of an interesting clinical case relating to the procedure of insertion of the bladder catheter performed by the nurse and a review of the literature, the authors explain the importance of the information that must be provided to the patient before carrying out any invasive procedure, even if not performed by the doctor". Given the unfortunate outcome that this case had (the only one analyzed, it is worth mentioning), namely with the death of the patient under analysis, 63 years old, I would suggest that the word "Interesting" be removed.

Dear reviewer n.1. Thank you very much for your review. We proceeded to remove the word "interesting".

In the Introduction, the authors refer to Italian legal guidelines on this context, but only in the "Discussion" part do they allude to the regulations in question, as well as the manifest insufficiencies. It would be appropriate to anticipate this legal context already in the "Introduction", but based on the main references.

Done, thank you very much.

The "Conclusions", however, seem to fall short of expectations, with only three paragraphs, the first of which is restricted to direct quotes. As there are seven co-authors, and after analyzing a clinical case that deserves care, especially because it urges that these conditions be the subject of increased vigilance, I would have expected a joint opinion from the authors on development strategies to be proposed and implemented for the benefit of patients and the clinical community. 

Dear reviewer we have improved our conclusions. Thanks again

Reviewer 2 Report

This paper concerns informed consent to medical treatment; the authors intend to focus on invasive procedures even if not performed by doctors.

While the attention to the topic and the conclusions presented by the authors in paragraph 5 are certainly appreciable, the article is not clearly structured and its scope is totally unclear. A case report is introduced but then it is not discussed in relation to the specific topic of consent. It is unclear what purpose the literature review presented in paragraphs 2 and 3 served, given that the literature identified is not adequately presented in paragraph 4 and refers to legal systems other than the Italian one. The presentation of the Italian context would presuppose an analysis of the Italian legal literature (it may be useful to consult the DoGi database: http://dati.igsg.cnr.it/dogi) and of the most recent case law of the Court of Cassation (e.g., Cass. civ. sez. III 12 May 2021, no. 12593).

Author Response

This paper concerns informed consent to medical treatment; the authors intend to focus on invasive procedures even if not performed by doctors.

While the attention to the topic and the conclusions presented by the authors in paragraph 5 are certainly appreciable, the article is not clearly structured and its scope is totally unclear. A case report is introduced but then it is not discussed in relation to the specific topic of consent. It is unclear what purpose the literature review presented in paragraphs 2 and 3 served, given that the literature identified is not adequately presented in paragraph 4 and refers to legal systems other than the Italian one. The presentation of the Italian context would presuppose an analysis of the Italian legal literature (it may be useful to consult the DoGi database: http://dati.igsg.cnr.it/dogi) and of the most recent case law of the Court of Cassation (e.g., Cass. civ. sez. III 12 May 2021, no. 12593).

Dear Reviewer, thanks for the important advice. We reviewed the whole article and checked the structure of the paper. We also studied the new sentences from the court of cassation. We have therefore modified the paper.

Round 2

Reviewer 2 Report

The paper has been revised and the new version can be accepted for publication.